# Sparkle: Mastering Basic Spatial Capabilities in Vision Language Models Elicits Generalization to Composite Spatial Reasoning

## Abstract

Vision language models (VLMs) have demonstrated impressive performance across a wide range of downstream tasks. However, their proficiency in spatial reasoning remains limited, despite its crucial role in tasks involving navigation and interaction with physical environments. Specifically, much of the spatial reasoning in these tasks occurs in two-dimensional (2D) environments, and our evaluation reveals that state-of-the-art VLMs frequently generate implausible and incorrect responses to composite spatial reasoning problems, including simple pathfinding tasks that humans can solve effortlessly at a glance. To address this, we explore an effective approach to enhance 2D spatial reasoning within VLMs by training the model on *basic spatial capabilities*. We begin by disentangling the key components of 2D spatial reasoning: direction comprehension, distance estimation, and localization. Our central hypothesis is that mastering these basic spatial capabilities can significantly enhance a model's performance on composite spatial tasks requiring advanced spatial understanding and combinatorial problem-solving. To investigate this hypothesis, we introduce *Sparkle*, a framework that fine-tunes VLMs on these three basic spatial capabilities by synthetic data generation and targeted supervision to form an instruction dataset for each capability. Our experiments demonstrate that VLMs fine-tuned with Sparkle achieve significant performance gains, not only in the basic tasks themselves but also in generalizing to composite and out-of-distribution spatial reasoning tasks (e.g., improving from 13.5% to 40.0% on the shortest path problem). These findings underscore the effectiveness of mastering basic spatial capabilities in enhancing composite spatial problem-solving, offering insights into systematic strategies for improving VLMs' spatial reasoning capabilities.

## 1 Introduction

Vision language models (VLMs) (OpenAI, 2023; Liu et al., 2023b; Chen et al., 2024c) have demonstrated near-human performance in tasks like image captioning (Chen et al., 2015), visual question answering (VQA) (Goyal et al., 2017; Singh et al., 2019) and abundant downstream tasks by combining visual and text inputs to reason about the physical world. However, these models exhibit significant limitations in understanding spatial relationships. For instance, as shown in Figure 1, state-of-the-art (SoTA) VLMs GPT-4o and InternVL2-Pro (OpenAI, 2023; Chen et al., 2024c) generate implausible responses to a shortest path problem that a human could solve at a glance, a simple 2D spatial reasoning task.

Nevertheless, 2D spatial reasoning is essential for VLMs to understand and interact with the physical environments, shaping their ability to solve mazes (Ivanitskiy et al., 2023; Wang et al., 2024), plan routes (Feng et al.,

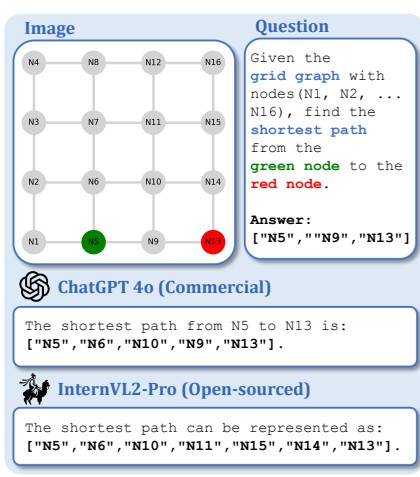

Figure 1: SoTA VLMs fail to solve the pathfinding problem, a simple 2D spatial reasoning task.

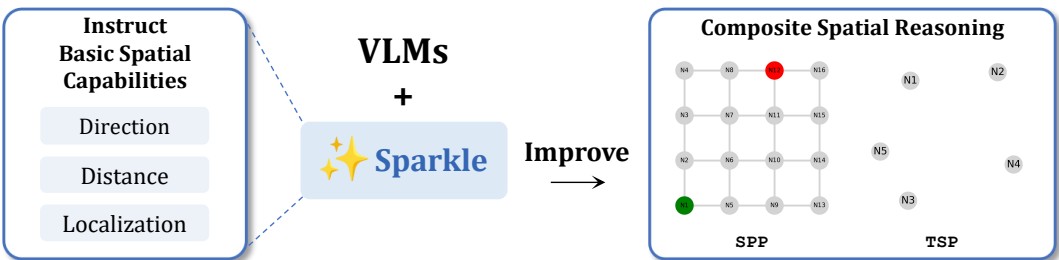

Figure 2: An overview of the workflow of the Sparkle framework.

2024; Chen et al., 2024b), and solve geometric problems like humans (Fernandes & de Oliveira, 2009). These tasks emphasize 2D spatial reasoning, requiring VLMs to process and navigate flat visual planes, interpret spatial relationships, and make decisions based on geometric understanding. Such capabilities are fundamental in translating visual input into actionable insights. While more and more VLMs are developed with larger training datasets and extensive benchmarks (Ge et al., 2024; Zhang et al., 2024), the focus on enhancing spatial reasoning has received comparatively less attention, despite its importance to the core capabilities of VLMs.

In this paper, we study VLMs' spatial reasoning capabilities in a 2D space by investigating three key questions: (1) How well do existing models perform on 2D spatial reasoning? (2) What fundamental tasks affect spatial reasoning capabilities in 2D? (3) Can mastering basic tasks help improve the performance of complex spatial reasoning?

We first analyzed various spatial reasoning tasks presented in existing works (Kamann & Rother, 2020), identifying the capabilities required for these tasks. From this analysis, we identified three basic capabilities fundamental for spatial reasoning in 2D space: direction comprehension, distance estimation, and localization. A systematic evaluation of the performance of existing open-source and closed-source VLMs on these three basic capabilities reveals that even the most advanced VLMs sometimes struggle with these fundamental tasks. For instance, in a simple 2D direction classification task, where a model is asked to determine the relative direction (top left, top right, bottom left, bottom right) of one object relative to another on a straightforward diagram with only two objects, the state-of-the-art VLM GPT-4o can achieve only 76.5% accuracy. In contrast, a human should be able to answer these questions correctly without much thought.

Most real-world spatial reasoning tasks, such as pathfinding (Lester, 2005; Cui & Shi, 2011), inherently require the composition of the basic capabilities identified above. A composite task is often subject to specific constraints that necessitate tailored solutions, unlike improving basic spatial reasoning capabilities, which can exhibit generalizability. In order to effectively improve the model's overall spatial reasoning capabilities in 2D space, we raise a conjecture: whether a VLM that masters the three basic capabilities can generalize and perform better on more complex composite spatial tasks. In other words, can a VLM exhibit compositional generalizability (van Zee, 2020) in spatial reasoning tasks?

To test this, we propose *Sparkle*, as shown in Figure 2, a framework that fine-tunes VLMs on these three basic spatial capabilities by programmatically generating synthetic data and providing supervision to form an instruction dataset for each capability. Our experimental results show that models trained on *Sparkle* achieve significant performance gains, not only in the basic tasks themselves (e.g., improving from 35% to 83% for InternVL2-8B on direction comprehension) but also in generalizing to composite and out-of-distribution general spatial reasoning tasks (e.g., improving from 13.5% to 40.0% on the shortest path problem). Additionally, our ablation study confirms the importance of mastering all three basic spatial reasoning capabilities. To summarize, our contributions are:

- We show that state-of-the-art VLMs struggle with composite spatial reasoning tasks that humans solve effortlessly.

- We identify three key components of spatial reasoning and construct an instruction-tuning method called Sparkle to improve these three fundamental spatial reasoning capabilities.

- Our experiments demonstrate that enhancing VLMs' basic spatial capabilities significantly improves their ability to generalize to out-of-distribution composite spatial tasks.

## 2 RELATED WORK

### 2.1 VISION LANGUAGE MODELS AND APPLICATIONS

Early works on VLMs, such as CLIP (Radford et al., 2021a) and ALIGN (Jia et al., 2021), leveraged contrastive learning to align visual and textual embeddings in a shared latent space, demonstrating strong capabilities in linking visual content with corresponding natural language descriptions. With the rapid advancement of Large Language Models (LLMs), modern VLMs increasingly combine pretrained vision models (Dosovitskiy et al., 2021; Chen et al., 2023) with powerful LLMs (Chiang et al., 2023; Bai et al., 2023a; Jiang et al., 2023; Cai et al., 2024) to facilitate a more cohesive understanding of both modalities (Liu et al., 2023b; Bai et al., 2023a; Chen et al., 2024c). This approach enables richer visual reasoning, open-ended image captioning, and more interactive multimodal dialogue systems.

VLMs have been applied in various pre-training tasks, such as image-text matching, masked image modeling, and multimodal reasoning (Li et al., 2022; 2023a; Wang et al., 2022b). In downstream tasks, they excel in applications like visual question answering (Antol et al., 2015; Wang et al., 2022a), image captioning (Li et al., 2020; Sidorov et al., 2020; Wang et al., 2021), image generation based on textual prompts (Ramesh et al., 2022; Baldridge et al., 2024), and aiding human-machine interactions in complex real-world settings, showcasing their versatility and potential across a broad range of vision language applications.

### 2.2 SPATIAL REASONING IN LLMS AND VLMS

Spatial reasoning in LLMs involves understanding and manipulating spatial relationships described in text. Early work focused on extracting spatial information from natural language (Hois & Kutz, 2011; Kordjamshidi et al., 2011). More recent efforts emphasize improving multi-hop spatial reasoning (Li et al., 2024b), especially in complex scenarios like 2D visual scenes (Shi et al., 2022). Key methods include pretraining on synthetic datasets to better capture spatial patterns (Mirzaee et al., 2021), and using in-context learning to generalize spatial reasoning across tasks, such as transforming spatial data into logical forms or visualizing reasoning traces (Yang et al., 2023b; Wu et al., 2024; Tang et al., 2024).

Building on these foundations, VLMs extend spatial reasoning by integrating visual inputs and often implicitly encode spatial knowledge through large-scale pretraining on visual-text datasets (Radford et al., 2021b; Li et al., 2023b). Early studies on VLMs primarily focus on understanding spatial relationships between objects in front-view images (Liu et al., 2023a), laying the groundwork for 2D spatial reasoning. More recently, research on VLMs has expanded to 3D reasoning tasks, which introduce additional challenges such as depth estimation (Chen et al., 2024a) and path planning (Chen et al., 2024b; Deng et al., 2020), as seen in applications like robotic grasping (Xu et al., 2023) and navigation (Shah et al., 2023; Chiang et al., 2024) in the embodied AI field (Li et al., 2024c). Despite these advances, 2D spatial reasoning remains more fundamental and flexible, as it can be applied to various tasks, including VQA (Ge et al., 2024; Kamath et al., 2023; Li et al., 2024a) and user interface grounding (Rozanova et al., 2021). Due to its broad applicability and foundational role, this work focuses on exploring 2D spatial reasoning capabilities within VLMs.

## 3 METHODOLOGY

In order to systematically evaluate and enhance the spatial reasoning capabilities of VLMs in 2D environments, we introduce the Sparkle framework, as illustrated in Figure 3. This section is structured as follows:

- Disentangling basic elements: How we identified the basic spatial capabilities of 2D spatial reasoning, and why these elements are foundational.
- Sparkle: We present the Sparkle framework to enhance VLMs' performances in 2D spatial reasoning by systematically improving the identified basic spatial capabilities.
- Tasks: We employ three spatial reasoning tasks specifically designed to evaluate both basic and composite spatial reasoning capabilities of VLMs.

Table 1: Overview of VLM tasks related to spatial reasoning and their required capabilities.

| Task | Recognition | Counting | Depth | Direction | Localization | Distance |
|---|---|---|---|---|---|---|
| QualSR (Freksa, 1991) | ✔ | ✘ | ✘ | ✔ | ✔ | ✔ |
| MapVQA (Wang et al., 2024) | ✔ | ✔ | ✘ | ✔ | ✔ | ✘ |
| NavVQA (Wang et al., 2024) | ✔ | ✔ | ✘ | ✔ | ✔ | ✘ |
| GridVQA (Wang et al., 2024) | ✔ | ✔ | ✘ | ✔ | ✔ | ✘ |
| VisualSR (Rajabi & Kosecka, 2023) | ✔ | ✘ | ✔ | ✔ | ✘ | ✔ |
| TvRecog. (Li et al., 2024a) | ✔ | ✘ | ✘ | ✘ | ✘ | ✘ |
| TvLoc. (Li et al., 2024a) | ✔ | ✘ | ✘ | ✘ | ✔ | ✘ |
| StaticSR (Li et al., 2024a) | ✔ | ✔ | ✘ | ✔ | ✘ | ✘ |
| DynamicSR (Li et al., 2024a) | ✔ | ✔ | ✘ | ✔ | ✘ | ✘ |
| SRR (Chen et al., 2022) | ✔ | ✘ | ✔ | ✘ | ✔ | ✔ |
| COCO-Spatial (Ranasinghe et al., 2024) | ✔ | ✘ | ✔ | ✔ | ✔ | ✘ |
| What's Up (Kamath et al., 2023) | ✔ | ✘ | ✔ | ✔ | ✘ | ✘ |
| Q-Spatial (Liao et al., 2024) | ✔ | ✘ | ✔ | ✘ | ✘ | ✔ |
| SpatialRGPT (Cheng et al., 2024) | ✔ | ✘ | ✔ | ✔ | ✔ | ✔ |

## 3.1 DISENTANGLING SPATIAL REASONING

To systematically disentangle basic 2D spatial reasoning capabilities, we first analyze the capabilities required in existing VLM benchmarks related to spatial reasoning, as shown in Table 1. While these benchmarks include a wide array of capabilities, such as image recognition and depth estimation, we narrow our focus to those most fundamental to 2D spatial reasoning. Depth estimation, though relevant to spatial reasoning, is more suited to 3D tasks and thus excluded from our analysis, as discussed in Section 2.2. We present the definitions of three basic components: (1) *Direction Comprehension*: The ability to understand the orientation of an object relative to a reference object; (2) *Distance Estimation*: The ability to gauge the magnitude of spatial displacement between objects; (3) *Localization*: The ability to determine the precise position of an object in space.

The selected basic spatial reasoning capabilities are foundational because they collectively represent the minimal components necessary to fully describe an object's position in 2D space. In particular, each of the three capabilities aligns with principles from Cartesian and polar coordinate systems, which serve as the mathematical bedrock for spatial representation: direction defines orientation, distance represents magnitude, and localization integrates both to precisely define an object's position, ensuring comprehensive spatial awareness (Zeng & Si, 2017). This decomposition enables a systematic evaluation of spatial reasoning by isolating the key dimensions of spatial understanding.

## 3.2 SPARKLE

To comprehensively investigate our hypothesis, we introduce Sparkle, a simple yet effective framework for constructing an instruction dataset focused on enhancing a model's spatial reasoning abilities. This framework only improves VLMs' basic spatial capabilities, and this design enables us to evaluate whether models that perform well on basic spatial reasoning tasks can also excel in more complex and composite problems.

### 3.2.1 INSTRUCTION DATA GENERATION

The design of our instruction dataset focuses on three basic spatial capabilities: direction, distance, and localization, based on insights provided in Section 3.1. The proposed fine-tuning pipeline does not require manual labeling, as all data can be programmatically generated.

We use $\mathbb{G}$ to denote a data generator that can generate a set of objects, $P = \{N_i\}_{i=1}^n$, representing a training sample of basic spatial capabilities. Each object $N_i = (x_i, y_i) \in \mathbb{R}^2$ consists of randomly sampled coordinates within a bounded region. For each basic capability $\mathcal{T} \in \{\text{dir.}, \text{dist.}, \text{loc.}\}$, we construct a dataset $D_\mathcal{T}$ containing input-output pairs $(\mathcal{X}^\mathcal{T}, \mathcal{Y}^\mathcal{T})$, where $\mathcal{X}^\mathcal{T}$ represents the inputs and $\mathcal{Y}^\mathcal{T}$ represents the corresponding ground truth outputs. Each input $\mathcal{X}^\mathcal{T}$ consists of: (1) A visual input $\mathcal{X}_V^\mathcal{T}$: A labeled diagram representing the spatial configuration of a sample of objects through a visual representation function $\mathbb{V}_\mathcal{T}(P)$, (2) A language prompt $\mathcal{X}_L^\mathcal{T}$: A question querying some aspects of the spatial properties for $P$.

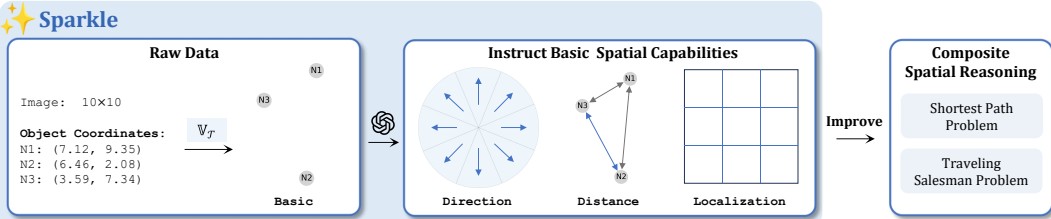

Figure 3: The proposed Sparkle framework.

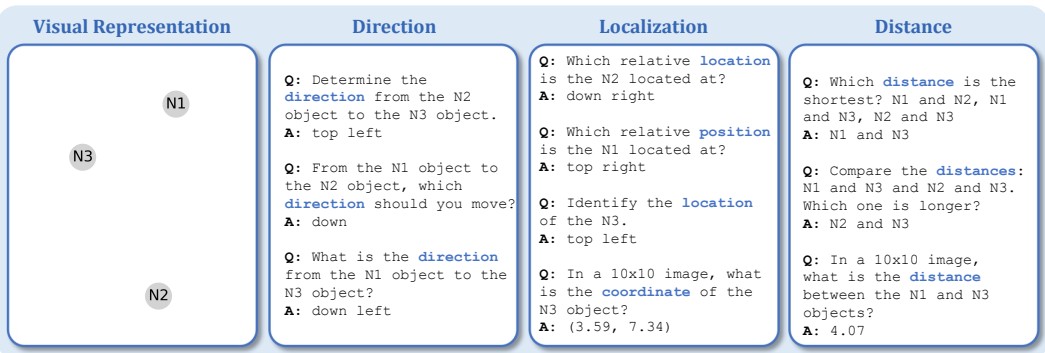

Figure 4: A data sample from the Sparkle dataset.

For example, to craft a training sample for direction comprehension, two objects, $N_1$ and $N_2$, are selected from $P$, and a question such as "What is the direction of $N_2$ relative to $N_1$?" is posed. The corresponding correct answer $Y^{\mathcal{T}}$ can be easily computed since we can access the exact coordinates of these objects, e.g., we can obtain the answer to the above question by calculating the vector from $N_1$ to $N_2$ based on their coordinates and map it to the corresponding directional label. Details about these generation processes can be found in Appendix §A.1.

The resulting training dataset consists of these generated questions and answers, paired with the corresponding visual representations, as shown in Figure 4. Specifically, the training pairs are represented as $\{(\mathcal{X}_L^{\text{train}}, \mathcal{X}_V^{\text{train}}, \mathcal{Y}^{\text{train}})\}$, where $\mathcal{X}_L^{\text{train}}$ represents the language-based queries, $\mathcal{X}_V^{\text{train}}$ represents the visual representations, and $\mathcal{Y}^{\text{train}}$ represents the corresponding answers. We also provide a complete data sample from the Sparkle training set in Appendix §A.5.1.

### 3.2.2 INSTRUCTION FINETUNING FOR BASIC TASKS

To enhance the spatial reasoning capabilities of VLMs, we use the Sparkle training set, denoted as $\mathcal{X}^{\text{train}} = \{(\mathcal{X}_L^{\text{train}}, \mathcal{X}_V^{\text{train}})\}$. The objective is to minimize the negative log-likelihood of the predicted answers. Specifically, the loss function $\mathcal{L}$ is defined as:

$$\mathcal{L}(\theta) = -\mathbb{E}_{(\mathcal{X}^{\text{train}}, \mathcal{Y}^{\text{train}})} \left[\log p(\mathcal{Y}^{\text{train}} \mid \mathcal{X}_V^{\text{train}}, \mathcal{X}_L^{\text{train}}; \theta)\right]$$

where $\theta$ represents the parameters of the VLM. The training aims to improve the model's proficiency in basic spatial reasoning tasks, which subsequently allows for an effective evaluation of its performance on more complex spatial challenges.

### 3.3 TASKS

The goal of the employed tasks is to evaluate the 2D spatial reasoning capabilities of VLMs and provide a foundation for studying how acquiring basic spatial capabilities can enhance performance on complex tasks. To achieve this, we follow key design criteria: (1) focus on spatial reasoning, and (2) progression from basic to composite tasks.

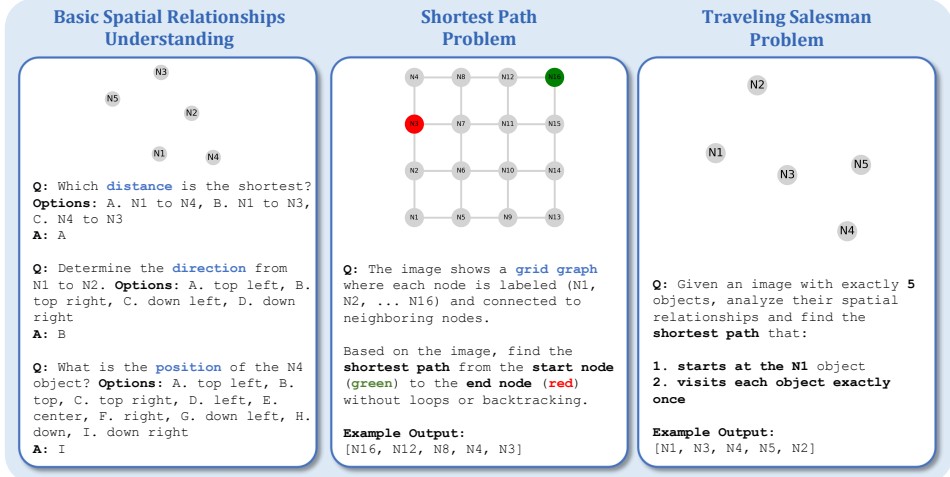

Figure 5: A sample from the evaluation dataset.

### 3.3.1 BASIC TASKS: ASSESSING FUNDAMENTAL SPATIAL CAPABILITIES

As shown in Figure 5 (left), the basic tasks in Sparkle are designed to assess the model's understanding of three basic spatial capabilities: (1) direction comprehension, (2) distance estimation, (3) localization.

In each basic task, the VLM is provided with an image containing several labeled data objects and a multiple-choice question about the spatial properties of these objects, with the goal of having the model answer these questions correctly. We first generate labeled diagrams that serve as visual inputs, then generate the questions (in multiple-choice format) and corresponding answer pairs, similar to the process described in Section 3.2, to obtain the basic task test set.

### 3.3.2 COMPOSITE TASKS: EVALUATING INTEGRATED SPATIAL REASONING

Building on the basic spatial relationships, the composite tasks introduce greater complexity. The objective here is to assess whether the model can apply basic spatial skills to solve problems requiring a combination of these skills or whether it has merely learned each skill in isolation without being able to generalize effectively. We choose the *Shortest Path Problem (SPP)* and the *Traveling Salesman Problem (TSP)* as composite tasks to evaluate the integration of basic capabilities.

**Shortest Path Problem (SPP)**  As shown in Figure 5 (middle), SPP evaluates the model's capability to compute the most efficient route between two objects on a 2D grid, requiring a combination of distance estimation and spatial planning.

Consider a grid $G$ of size $n \times n$, with two special objects: the start object $N_{\text{start}} = (x_s, y_s)$ and the end object $N_{\text{end}} = (x_e, y_e)$. We employ a language model LM generates the prompt $\mathcal{X}_L^{\text{spp}}$ using a predefined prompt template $\mathbb{P}_{\text{spp}}$, expressed as: $\mathcal{X}_L^{\text{spp}} = \text{LM}(\mathbb{P}_{\text{spp}}(G, N_{\text{start}}, N_{\text{end}}))$. The visual input is produced similar to basic tasks: $\mathcal{X}_V^{\text{spp}} = \mathbb{V}_{\text{spp}}(G, N_{\text{start}}, N_{\text{end}})$.

The combined input for the VLM is $\mathcal{X}^{\text{spp}} = (\mathcal{X}_V^{\text{spp}}, \mathcal{X}_L^{\text{spp}})$, and the model is expected to predict the shortest path $\widehat{\mathcal{Y}}^{\text{spp}}$, which is evaluated against the true shortest path, $\mathcal{Y}^{\text{spp}}$, computed using standard algorithms.

**Traveling Salesman Problem (TSP)**  As shown in Figure 5 (right), the TSP represents a more challenging spatial reasoning task, involving combinatorial optimization. The model must find the shortest possible route that visits each object exactly once and returns to the starting object.

Given $n$ objects $P^{\text{tsp}} = \{N_i\}_{i=1}^n$ sampled from $\mathbb{G}$, the ground truth solution $\mathcal{Y}^{\text{tsp}}$ is computed using a TSP solver $\mathbb{M}_{\text{tsp}}(P^{\text{tsp}})$. Similarly, the input to VLMs consists of a visual representation $\mathcal{X}_V^{\text{tsp}} = \mathbb{V}_{\text{tsp}}(P^{\text{tsp}})$ and a corresponding language prompt $\mathcal{X}_L^{\text{tsp}}$. The complete input query is $\mathcal{X}^{\text{tsp}} =$

$(\mathcal{X}_V^{\text{tsp}}, \mathcal{X}_L^{\text{tsp}})$. Similarly, the model's predicted order of visiting all objects, $\widehat{\mathcal{Y}}^{\text{tsp}}$, is then evaluated against the ground truth solution $\mathcal{Y}^{\text{tsp}}$.

### 3.3.3 DISCUSSION

Given that the SPP can be solved in polynomial time, we expect that if the model can effectively combine its knowledge of basic spatial concepts, it will show significant improvements in solving this task efficiently. On the other hand, the TSP is an NP-hard problem, requiring combinatorial optimization to obtain the exact solution. We include the TSP to push the limits of the model's spatial reasoning capabilities, aiming to investigate how well the model can manage more complex problem-solving tasks beyond the basic integration of spatial skills.

## 4 EXPERIMENTS

In this section, we provide our findings along with the supporting results to demonstrate the effectiveness of the Sparkle framework. Specifically, the experiments are designed to answer the following research questions: **RQ1**: Can mastering basic 2D spatial components enhance overall spatial reasoning capability in VLMs? **RQ2**: What insights from the results of evaluations (Section 4.2.1), enhancements (Section 4.2.2), and spatial components (Section 4.3) can guide improvements in model design, training strategies, and data collection for spatial reasoning in VLMs?

### 4.1 SETTINGS

**Models** We tested open-source and commercial models to evaluate and enhance VLMs' spatial reasoning capabilities. For commercial VLMs, we used GPT-4o from OpenAI (Yang et al., 2023a) and Google-Gemini (GeminiTeam et al., 2023). We included LLaVA1.6 (Liu et al., 2024) and InternVL2 (Chen et al., 2024c) for open-source models. Detailed model specifications and configurations are provided in Appendix §A.1. We use the MS-Swift library (Zhao et al., 2024) and apply the LoRA (Hu et al., 2022) fine-tuning strategy, with low-rank dimension of 32. We set a constant learning rate of 1e-4 and a batch size of 1. All training and evaluation tasks are performed on GPU clusters with 8×NVIDIA A100 machines. Further details can be found in Appendix §A.1.

**Data** We built the Sparkle training dataset by generating 2,000 images. each with 17 instruction-answer pairs that describe the spatial relationships between objects, resulting in 34K samples in total. Out of these 17 pairs, 3 focus on directions between objects, 7 on distances (including 4 for comparing distances and 3 for estimating numerical distances), and 6 relate to localization (with 3 for identifying object locations and 3 for estimating exact positions). The final instruction describes the overall spatial relationships in the image. This setup helps ensure the VLM maintains its capability to follow instructions effectively. Our evaluation includes tasks of: (1) shortest path problem (SPP), (2) traveling salesman problem (TSP), and (3) basic spatial relationship understanding. For each of them, we generated 200 samples, which together make up the evaluation set. For SPP and TSP, we use LLaMA 3.1 (Dubey et al., 2024) to process the VLMs' responses into list formats to enable metric computation. For the basic tasks, we structured them as a multiple-choice question format. In addition, for SPP and TSP, we designed experiments that vary by grid size and the number of objects involved. Detailed data statistics and sample data are provided in Appendix §A.2.

To further assess the generalizability of the improved spatial reasoning capabilities, we evaluated VLMs on existing general spatial reasoning-related benchmarks to examine its out-of-distribution performance. The general spatial benchmarks we used include What's Up, COCO-spatial, and GQA-spatial (Kamath et al., 2023), which feature real-world images and spatial reasoning questions.

**Metrics** For most of the tasks, we report accuracy as the primary evaluation metric. However, for tasks like SPP and TSP, where distance is crucial, we also use an additional metric *Margin* as a complementing one. This metric measures the extent to which the total distance of the solved path exceeds the optimal path (i.e., the shortest), expressed as a ratio of their summed distances. A lower Margin indicates better performance. Formally, the Margin is defined as: Margin $= \sum(\text{dist}(\text{solved\_path}) - \text{dist}(\text{optimal\_path})) / \sum \text{dist}(\text{optimal\_path})$, where the function $\text{dist}(\cdot)$ computes the total distance of a given path.

Table 2: VLMs' performance on basic and composite spatial reasoning tasks. "SPP-$n$Grid" denotes shortest path problem on $n \times n$ grids. "TSP-$n$Obj" denotes traveling salesman problem with $n$ objects.

| Model | Basic Tasks | | | SPP-4Grid | | SPP-5Grid | | TSP-4Obj | | TSP-5Obj | |
|---|---|---|---|---|---|---|---|---|---|---|---|
| | Loc. | Dist. | Dir. | Acc↑ | Margin ↓ | Acc↑ | Margin↓ | Acc↑ | Margin↓ | Acc↑ | Margin↓ |
| GPT-4o | 67.5 | 41.5 | 76.5 | 74.5 | 0.089 | 78.5 | 0.178 | 23.5 | 0.001 | 21.5 | 0.195 |
| Gemini | 61.5 | 40.5 | 55.0 | 67.0 | 0.208 | 65.0 | 0.188 | 11.5 | 0.023 | 21.7 | 0.070 |
| LLaVA1.6-7B | 24.5 | 37.0 | 30.5 | 1.5 | – | 0.0 | – | 16.0 | 0.132 | 5.0 | 0.476 |
| InternVL2-26B | 62.5 | 45.5 | 58.0 | 15.5 | 0.5 | 10.9 | 1.135 | 21.5 | 0.074 | 12.5 | 0.265 |
| InternVL2-8B | 60.5 | 44.5 | 35.0 | 16.5 | 3.893 | 13.5 | 1.127 | 17.5 | 0.073 | 11.5 | 0.302 |
| + Sparkle-Instruct | 73.0 | 84.0 | 83.0 | 36.5 | 0.571 | 40.0 | 0.466 | 20.0 | 0.043 | 14.5 | 0.239 |
| Δ | +21% | +89% | +137% | +121% | -85% | +196% | -58% | +14% | -41% | +26% | -21% |

## 4.2 MAIN RESULTS

### 4.2.1 EVALUATION OF EXISTING VLMS

From Table 2, we observe that even the state-of-the-art commercial VLMs cannot obtain satisfactory results on composite tasks like SPP and TSP. Open-source models achieve even worse performance ($\leq$25% accuracy) on these tasks. Specifically, LLaVA perform poorly particularly on SPP compared to TSP, which may attribute to the grid data structure in SPP is more complex for VLMs to perceive compared to handling just a few objects in TSP, indicating that these VLMs struggle with visual representations involving intricate spatial structures. Performance on the TSP task worsens as the number of objects increases across most models, highlighting the growing difficulty of spatial reasoning with more objects. However, in SPP, increasing the grid size has little impact on performance, indicating that a larger grid does not increase the difficulty of reasoning. This result aligns with our initial design principles, where SPP was intended to combine basic spatial relationship understanding with a straightforward form of spatial planning.

To delve into how VLMs behave poorly on the composite spatial reasoning tasks, we further examine their performance on basic spatial relationship understanding, i.e. direction, location and localization comprehension. As shown in Table 2, even the state-of-the-art VLM GPT-4o struggles with basic spatial relationship understanding, achieving only 67.5%, 41.5%, and 76.5% accuracy on the direction, distance, and localization tasks, respectively. This investigation helps explain why VLMs underperform on composite tasks, as their inadequate basic spatial reasoning capabilities directly hinder their ability to handle more complex spatial challenges.

### 4.2.2 EFFECTIVENESS OF SPARKLE

To demonstrate the effectiveness of Sparkle, we present results from fine-tuning InternVL2-8B with this method. The results reveal significant improvements in both basic and composite tasks, indicating that 2D spatial reasoning capabilities can be significantly improved when a model effectively masters the basic components of 2D spatial reasoning.

Specifically, Sparkle only contains data for basic spatial relationship understanding. However, after fine-tuning with this data, VLMs improved in basic spatial reasoning (around 80%) and showed significant gains (around 90%) in composite tasks. This justifies the soundness of our abstraction of spatial reasoning in 2D space into three basic components (i.e., localization, distance, and direction), and that improving these basic capabilities could effectively enhance VLMs' overall spatial reasoning, enabling it to tackle more complex tasks. This finding highlights the potential of strengthening basic capabilities to improve problem-solving performance in VLMs. When comparing the improvements of the InternVL2 model on SPP and TSP, we observe that the gains (around 20%) on TSP are much smaller than those on the SPP task (160%). One possible explanation is that the TSP involves more complex optimization challenges, which may not be as easily addressed by simply improving basic spatial reasoning skills, as discussed in Section 3.3.3. This underscores the need for further research into the optimization capabilities of language models, a topic we hope our findings will inspire. Additionally, we present results of Sparkle on Qwen-VL-7B in Appendix §A.3.1, demonstrating its effectiveness across various different VLMs.

Table 3: Results on general spatial tasks.

| Model | What's Up | COCO-Spatial | | GQA-Spatial | |
|---|---|---|---|---|---|
| | | 1Obj | 2Obj | 1Obj | 2Obj |
| GPT-4o | 95.9 | 88.2 | 49.7 | 89.4 | 63.6 |
| Gemini | 69.4 | 50.8 | 34.1 | 42.9 | 21.7 |
| LLaVA1.6-7B | 44.9 | 14.4 | 6.0 | 12.6 | 2.2 |
| InternVL2-26B | 87.9 | 72.7 | 62.9 | 91.4 | 75.0 |
| InternVL2-8B | 92.7 | 92.5 | 71.3 | 97.5 | 85.3 |
| + Sparkle-Instruct | 93.9 | 93.0 | 78.4 | 98.0 | 90.0 |
| Δ | +1.3% | +0.5% | +10% | +0.5% | +5.5% |

### 4.2.3 GENERALIZABILITY

In the previous subsection, we have shown that spatial reasoning improvements can generalize from simple tasks to more complex ones. In this section, we evaluate this generalization further by testing spatial reasoning performance in an out-of-distribution (OOD) visual representation setting.

Specifically, we investigate whether the enhanced spatial reasoning capabilities transfer to other general VLM spatial tasks. As seen in Table 3, there are consistent gains across general VLM benchmarks related to spatial reasoning. For instance, the COCO-spatial and GQA-spatial benchmarks illustrate that current VLMs often struggle to accurately capture spatial relationships between two objects. However, with our Sparkle framework, this capability is greatly improved.

These findings suggest that future work designing and training VLMs should consider improving spatial reasoning of VLMs by decomposing into basic capabilities to enhance the general performance. Our results demonstrate that the Sparkle framework is simple and highly effective in enhancing spatial reasoning capabilities in VLMs.

### 4.3 ABLATION STUDIES

In this section, we present the results of ablation studies on the proposed Sparkle framework, using the InternVL2-8B model for demonstration.

### 4.3.1 IMPACT OF TRAINING COMPONENTS

To evaluate the impact of different training components, we compared Sparkle to several variants. First, we trained InternVL2-8B on individual spatial reasoning tasks with our Sparkle framework, resulting in *Sparkle(Direction, Distance, Localization)*. Additionally, we tested a version called *Sparkle w/o Num* that excludes numerical information (i.e., distance and location estimation) in Sparkle. All of the four variants are trained with the same number of total samples as the full Sparkle model. The results shown in Figure 6 reveal two key insights: First, *Sparkle w/o Num* consistently underperforms compared to the full Sparkle model, particularly in tasks that require strong distance reasoning, such as TSP. This suggests that incorporating numerical information dur-

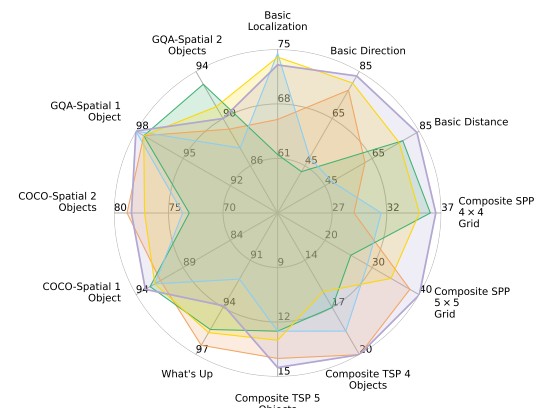

Figure 6: Ablation results showing accuracy for different Sparkle variants: Sparkle ▇; Sparkle without numerical information ▇; Sparkle (Localization) ▇; Sparkle (Distance) ▇; Sparkle (Direction) ▇.

ing training significantly enhances the model's capability in tasks involving distance reasoning and other related composite challenges. Second, training on specific spatial reasoning subsets can sometimes yield optimal performance for certain tasks. For example, *Sparkle (Direction)* achieves 96.4% accuracy on the What's Up benchmark, indicating that task-specific training can be highly effective. This highlights the importance of tailoring the training process to the unique characteristics of individual tasks. When a task emphasizes a particular spatial reasoning capability, focusing the training data on that aspect can improve performance on the targeted task. Overall, the full Sparkle

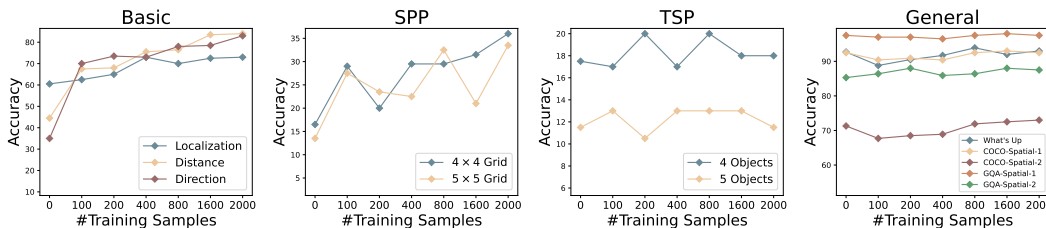

Table 4: Results of Sparkle on InternVL2-8B with varying training sample sizes.

framework consistently delivers the best results across the majority of benchmarks, demonstrating the effectiveness of a more comprehensive approach to training.

### 4.3.2 IMPACT OF TRAINING SAMPLE SIZE

In this section, we varied the training sample size in Sparkle and evaluated its impact on spatial reasoning tasks. The results are shown in Figure 4. Several key trends emerge from the results. First, we observe a general improvement in VLM performance as the training sample size increases despite some fluctuations in the curve. However, a noteworthy finding is the existence of task-specific sweet spots, beyond which performance gains taper off or degrade. This suggests that scaling up training samples does not always yield proportional improvements. For example, in the TSP task, performance begins to degrade once the number of training samples surpasses a threshold (around 800). This is likely because, after mastering basic spatial relationships, the model may focus on locally optimal choices, such as selecting the nearest objects to form a path, rather than optimizing the entire path. As the model grows more confident in these local decisions, it may sacrifice the global optimality of the solution, resulting in suboptimal performance.

### 4.4 DISCUSSION

The analysis and results confirm that mastering basic 2D spatial reasoning capabilities through Sparkle can significantly enhance VLMs' overall spatial reasoning in composite tasks (e.g., spatial planning) and general spatial tasks. This directly addresses RQ1 and supports the assumption presented in the methods section.

Turning to RQ2, the evaluation results revealed the limitations of existing VLMs, particularly in their capability to perceive complex spatial structures, as evidenced in tasks like SPP. This highlights the need for improved model and training designs to support more detailed spatial reasoning. Moreover, introducing synthetic data focusing on basic spatial relationships has proven to enhance overall VLM spatial reasoning performance, offering a clear path for future spatial data collection. Lastly, our ablation study suggests that training specific spatial reasoning capabilities in isolation yields the best results for tasks that demand focused spatial abilities. Therefore, in terms of training strategy, our findings suggest adopting a pre-train and fine-tune approach (i.e., using diverse spatial data in pretraining and fine-tuning specific spatial capabilities tailored to particular tasks) to improve VLMs' performances on corresponding tasks.

## 5 CONCLUSION

This work presents the Sparkle framework to address the relatively limited spatial reasoning ability of Vision Language Models (VLMs). Sparkle is designed to enhance spatial reasoning by focusing on three fundamental capabilities: direction comprehension, distance estimation, and localization. Our experiments demonstrate that fine-tuning on these basic capabilities leads to substantial improvements not only in the basic tasks but also in more complex, composite spatial reasoning challenges, thereby showcasing the compositional generalizability of our method. Furthermore, our analysis confirms that mastering all three basic spatial reasoning capabilities is essential for broader generalization, ultimately strengthening VLMs' ability to interact with the physical world.

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

# A APPENDIX

## A.1 IMPLEMENTATION DETAILS

In addition to the experimental settings outlined in Section 4.1, we provide the following categorized implementation details for this work.

For model specifications, the GPT-4o model used in our experiments and demonstrations is based on the gpt-4o-2024-05-13 version, while the Gemini model is Gemini 1.5 Flash. For TSP data generation, we used an open-source Python TSP solver[1] to obtain the ground truth visiting order of the given object coordinates.

For VLM evaluations, we focused on four directional categories (top left, top right, bottom left, and bottom right) to make it easier for VLMs to distinguish between directions. To discretize object locations for localization learning in VLM, the 2D space is proportionally divided using 40% and 60% thresholds along both the $x$ and $y$ axes, creating nine distinct regions (center, top, bottom, left, right, top-left, top-right, bottom-left, bottom-right). Detailed data statistics and distribution visualizations are provided in Section A.2.

To extract and format the VLMs' responses, we used the LLaMA 3.1 language model (Dubey et al., 2024), which converts the results into the required format for metric calculations. The specific prompts used for each task are detailed in Section A.4. The evaluation for basic spatial relationship understanding is intuitive, as it follows a multiple-choice question format. For the SPP evaluation, we check two criteria: (1) whether the solved path is valid on the grid, and (2) whether the length of the solved path is indeed the shortest between the given start and end objects. For the TSP evaluation, a path is considered "correct" only if it exactly matches the solution from the TSP solver mentioned above. To reduce the difficulty for VLMs in solving TSP, we explicitly specify the starting object in our implementation.

## A.2 DATA STATISTICS

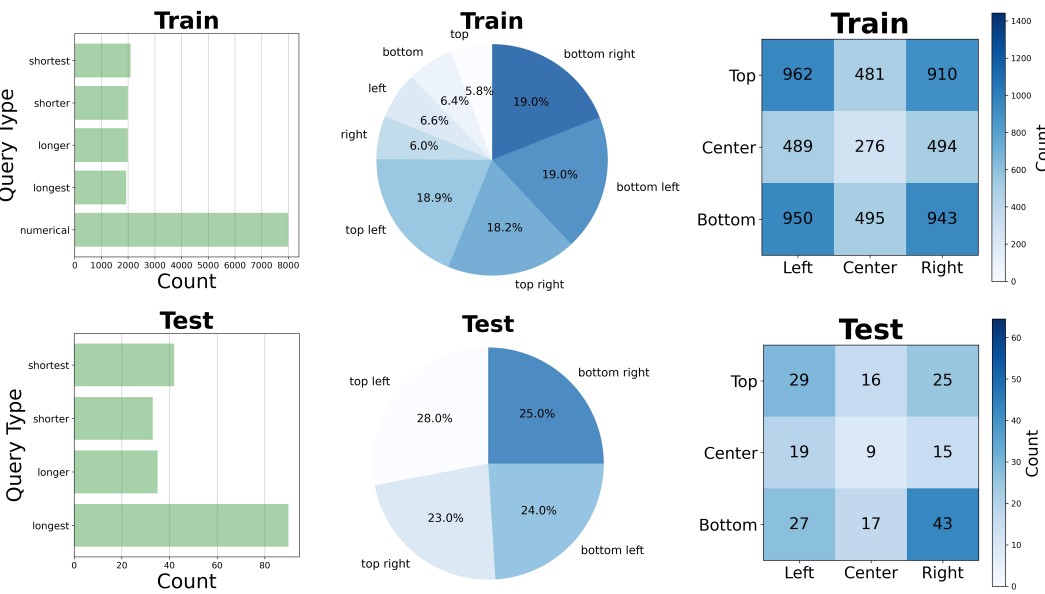

Figure 7: Data statistics of basic spatial relationships (from left to right: distance, direction, and localization statistics).

To complement Section 4.1, this section provides detailed statistics of the data from Sparkle training set and evaluation. We begin by discussing data related to basic spatial relationships (i.e., distance,

---

[1]https://github.com/fillipe-gsm/python-tsp

direction, localization), covering both Sparkle training set and the spatial relationship understanding task in the evaluation set.

Figure 7 illustrates various statistics. In the left column, we see the distribution of questions and instructions related to the *distance* between objects, which includes comparative expressions (e.g., shortest, shorter, longer, longest) and numerical distance estimations considered only in Sparkle training set. The training set shows a fairly even distribution of comparison queries, while in the test set, queries involving the "shortest" and "longest" distances occur more frequently than those involving "shorter" and "longer".

The middle column of Figure 7 presents the data concerning *directional* relationships between objects. We divided the 2D space into direction sectors: four sectors for testing and eight for training. The directional relationships of "bottom-right", "bottom-left", "top-right", and "top-left" each make up about 19% of the training data, while "top", "bottom", "left", and "right" each account for roughly 6%. In the test set, the four main directional relationships are distributed evenly.

Lastly, the right column in Figure 7 shows the *localization* data. Objects are most frequently located in the corners of the space (i.e., top-left, top-right, bottom-left, and bottom-right) in both the training and test sets. The number of objects placed in "top", "bottom", "left", and "right" positions is about half that of those in the corners, while the fewest objects are placed in the center. This is due to the intentional narrowing of the center area as we explained in Section A.1, which reduces the likelihood of randomly generated objects being placed there. Since there is no clear distinction between regions like "left" and "top-left", this narrowed design encourages VLMs to accurately distinguish specific areas such as the "center", "top", "bottom", "left", and "right"

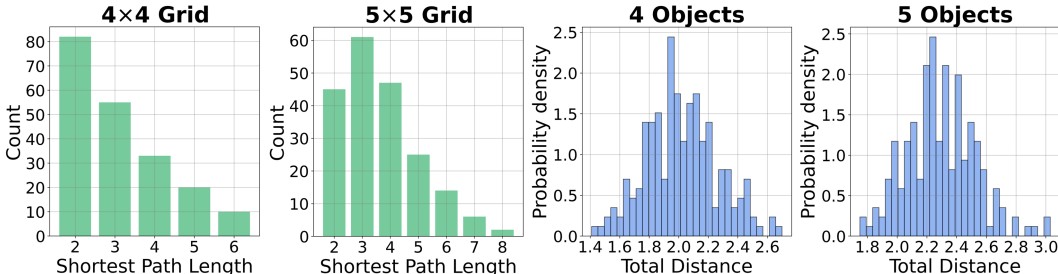

Figure 8: Data statistics of composite spatial reasoning tasks in the evaluation set.

Figure 8 presents data statistics for composite spatial reasoning tasks. The two left subfigures show the distribution of ground truth shortest path lengths in $4 \times 4$ and $5 \times 5$ grids, while the two right subfigures depict the distribution of total distances for the optimal path in the TSP with 4 and 5 objects.

### A.3 ADDITIONAL EXPERIMENTS

#### A.3.1 QWEN-VL EXPERIMENTS

| Model | Spatial Relationships | | | Traveling Salesman Problem | | | | COCO-Spatial | | GQA-Spatial | | What's |
|---|---|---|---|---|---|---|---|---|---|---|---|---|
| | Loc. | Dist. | Dir. | 4 Objects | | 5 Objects | | 1 Object | 2 Objects | 1 Object | 2 Objects | Up |
| | Acc. | Acc. | Acc. | Acc. | Marg. | Acc. | Marg. | Acc. | Acc. | Acc. | Acc. | Acc. |
| Qwen-VL-7B | 22.0 | 37.0 | 24.5 | 10.5 | 0.126 | 2.5 | 0.458 | 89.8 | 74.3 | 98.5 | 94.0 | 42.7 |
| + Sparkle-Instruct | 56.0 | 54.5 | 61.0 | 16.5 | 0.069 | 9.0 | 0.367 | 96.3 | 86.8 | 98.5 | 96.2 | 48.1 |
| Δ | +155% | +47% | +149% | +57% | -45% | +260% | -20% | +7% | +17% | − | +2% | +13% |

Table 5: Results of Qwen-VL Enhanced with Sparkle-Instruct

To further validate the generalizability of Sparkle, we conducted experiments using Qwen-VL (Bai et al., 2023b). The results, presented in Table 5, show significant improvements after fine-tuning with Sparkle compared to the original InternVL2-8B model.

Specifically, there is an approximately 120% improvement in the basic spatial relationship understanding task, a 150% improvement in the accuracy of composite tasks, and an 8% improvement in general spatial tasks. However, we excluded the results of the SPP task from our analysis, as the original performance of Qwen-VL-7B was too poor to allow for insightful comparison. This underperformance is not primarily due to limitations in spatial reasoning, but rather issues with visual recognition capabilities, as discussed in Section 4.2.1.

#### A.3.2 ABLATION STUDY RESULTS ON MARGIN METRIC

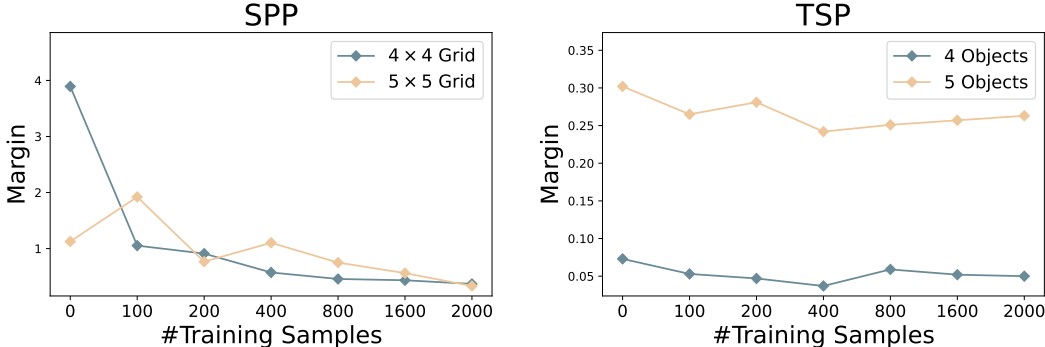

Figure 9: Margin results of InternVL2-8B with varying training sample sizes.

We have included the ablation study results on the margin metric here. As shown in Figure 9, a similar trend can be observed, consistent with the analysis discussed in Section 4.3.

## A.4 Prompts for Extracting Inference Results from VLMs

In this section, we provide the designed prompts for a LM to extract results from VLMs' responses.

### A.4.1 Multi-choice Questions

> **Prompt for Extracting Results from VLMs' Responses to Multiple-Choice Questions**
>
> ```
> Extract the option capital letter from the result and return it as \\boxed{{X}}, where
> X is the letter.
>
> Provide no additional content. The result is: ```{result}```.
>
> Make sure your response is in the \\boxed{{X}} format.
> ```

The above prompt is adopted for all evaluations that in a Multi-choice Questions format.

### A.4.2 Shortest Path Problem

> **Prompt for Extracting Results from VLMs' Responses to Shortest Path Problems**
>
> ```
> Extract the sequence of node labels from the given input and return it as a Python
> list.
>
> **Return Format:**
> - Do not include any additional text or explanations.
> - Ensure that the response is a single list containing only the node text labels (N1,
> N2, ...).
> - If no valid action sequence is found, return 'None'.
>
> **Example Output format:**
> ```
> [node1 text label, node2 text label, ...]
> ```
>
> Now, extract the result from the following input: ```{result}```. Strictly adhere to
> the return format.
> ```

### A.4.3 Traveling Salesman Problem

> **Prompt for Extracting Results from VLMs' Responses to Traveling Salesman Problems**
>
> ```
> Extract the sequence of movements from the given input and return it as a Python list
> of object names.
>
> **Return Format:**
> - Do not include any additional text or explanations.
> - Ensure that the response is a single list containing only the object names.
>
> **Expected Output Format:**
> ```
> {output_format}
> ```
>
> Now, extract the result from the following response: ```{result}```. Strictly adhere to
> the output format.
> ```

## A.5 SAMPLE DATA DEMONSTRATION

In this part, we provide detailed data sample from our experiments.

### A.5.1 DATA SAMPLE FROM SPARKLE TRAINING SET AND EVALUATION

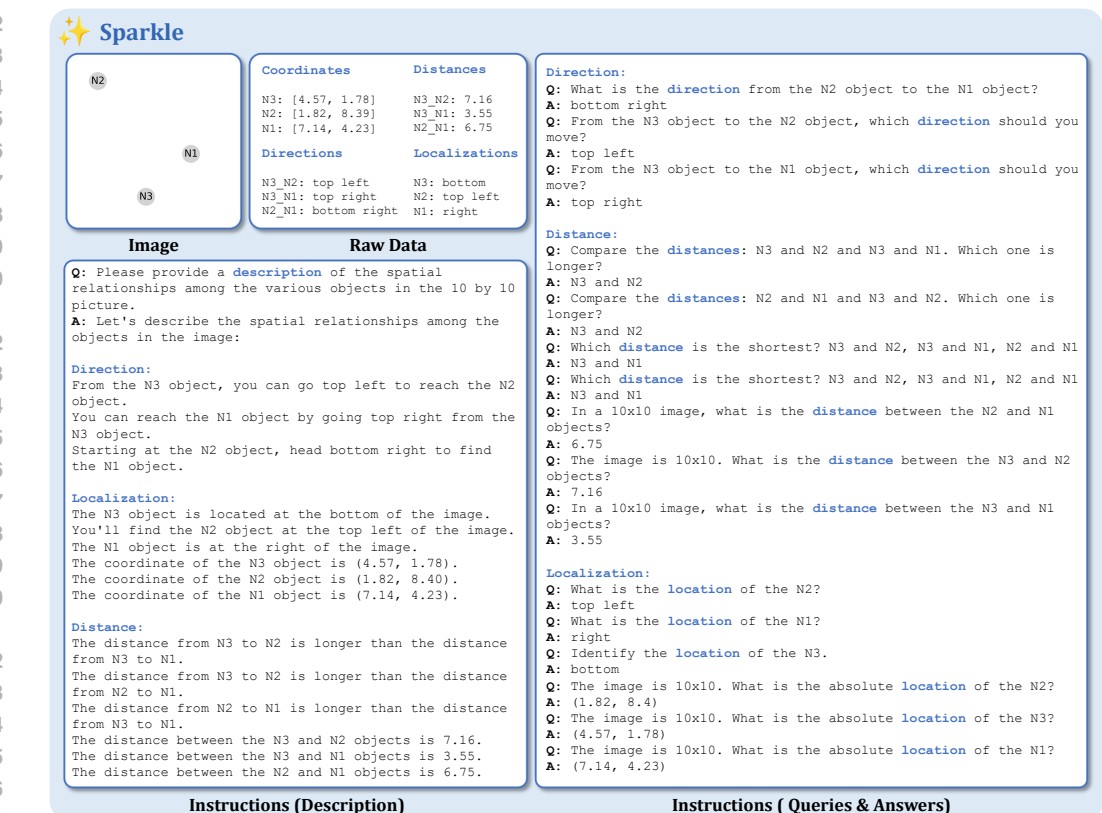

Figure 10: A data sample from the Sparkle training set.

### A.5.2 DATA SAMPLE FROM THE BASIC SPATIAL RELATIONSHIP UNDERSTANDING TASK

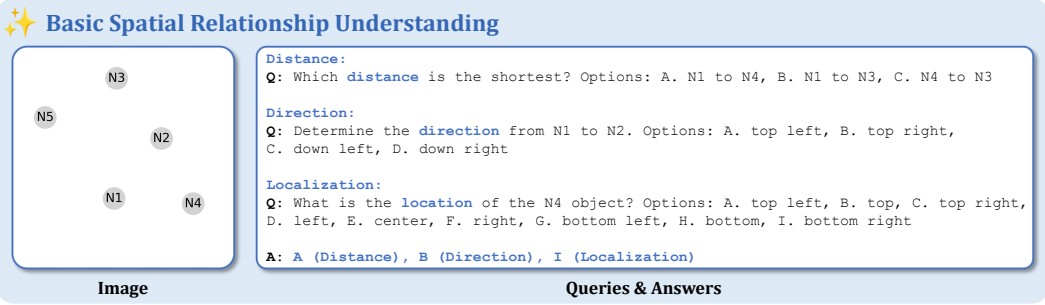

Figure 11: A data sample for Basic Spatial Relationship Understanding

### A.5.3 DATA SAMPLE FROM THE SHORTEST PATH PROBLEM

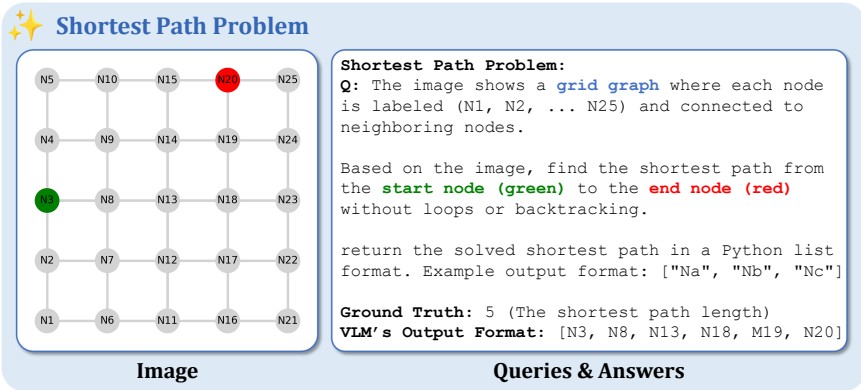

Figure 12: A data sample from the Shortest Path Problem.

### A.5.4 DATA SAMPLE FROM THE TRAVELING SALESMAN PROBLEM

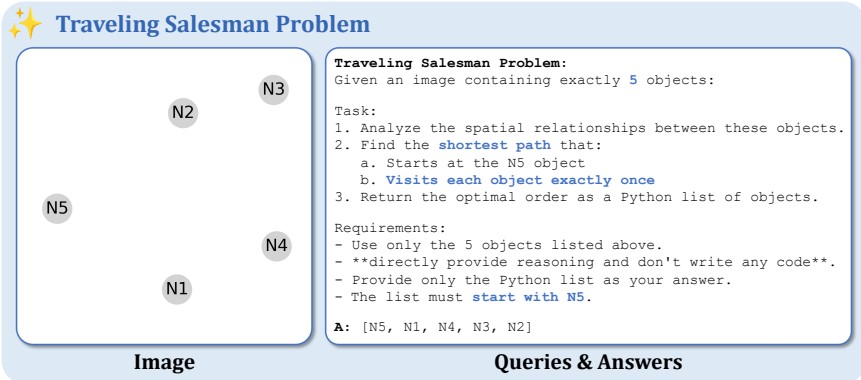

Figure 13: A data sample from the Traveling Salesman Problem.

### A.5.5 DATA SAMPLE FROM GENERAL SPATIAL TASKS

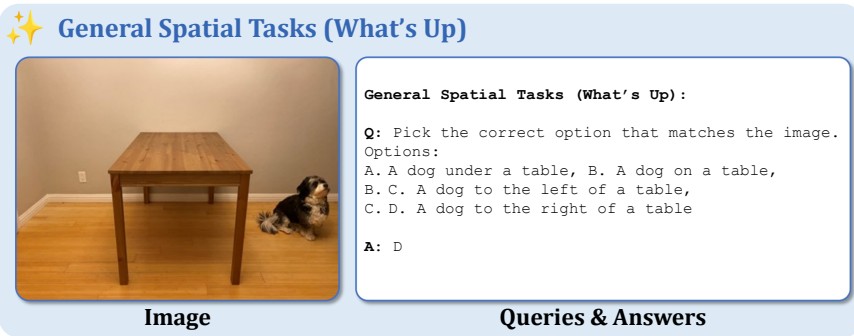

Figure 14: A data sample from the General Spatial Tasks ("What's Up").

## A.6 SCREENSHOTS OF CHAT WITH VLMS

### A.6.1 GPT-4O

Figure 15: Screenshot supporting Figure 1: Chat interactions with GPT-4o.

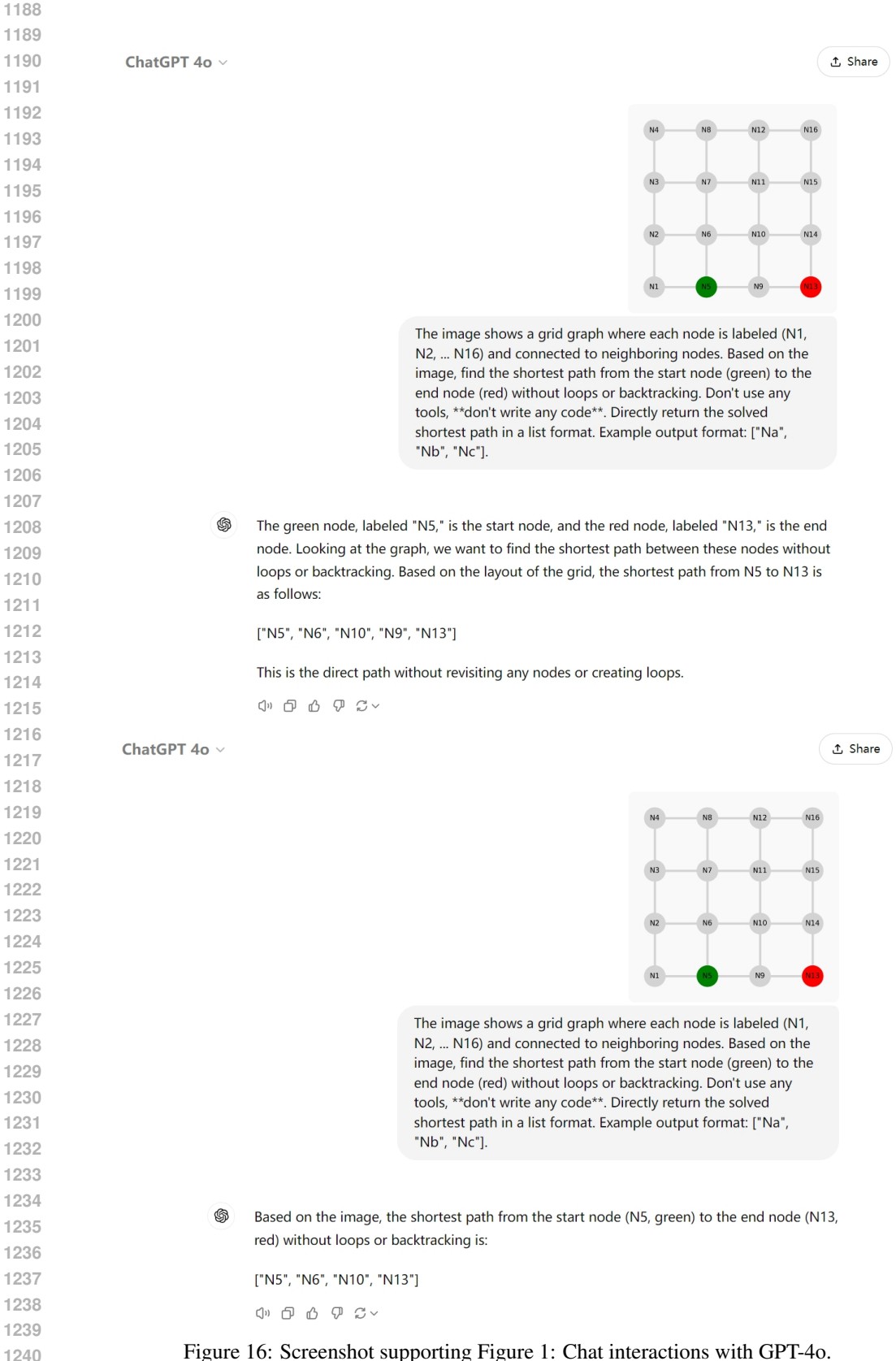

Figure 16: Screenshot supporting Figure 1: Chat interactions with GPT-4o.

### A.6.2 INTERNVL2-PRO

# InternVL2

Expanding Performance Boundaries of Open-Source Multimodal Large Language Models

> The image shows a grid graph where each node is labeled (N1, N2, ... N16) and connected to neighboring nodes. Based on the image, find the shortest path from the start node (green) to the end node (red) without loops or backtracking. Don't use any tools, **don't write any code**. Directly return the solved shortest path in a list format. Example output format: ["Na", "Nb", "Nc"].

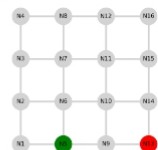

> (In this conversation, 1 image was uploaded, 1 image in total)

> To determine the shortest path from the green start node (N5) to the red end node (N13), we need to traverse through the graph using only adjacent nodes.
>
> Here's one possible path:
>
> - Start at N5
> - Move right to N6
> - Move down to N10
> - Move right to N11
> - Move down to N15
> - Move right to N14
> - Move up to N13
>
> Therefore, the shortest path can be represented as:
>
> ```
> ["N5", "N6", "N10", "N11", "N15", "N14", "N13"]
> ```
>
> This path ensures that no node is visited more than once and reaches the destination efficiently.

Clear Chat History    Regenerate

Figure 17: Screenshot supporting Figure 1: Chat interactions with InternVL2-Pro.

