# OpenReview forum: "Sparkle: Mastering Basic Spatial Capabilities in Vision Language Models Elicits Generalization to Composite Spatial Reasoning"
_ICLR.cc/2025/Conference — ICLR 2025 Conference Withdrawn Submission_

### Official Review · Reviewer_6ZbR · 2024-11-02

**Soundness:** 2
**Presentation:** 3
**Contribution:** 2
**Rating:** 5
**Confidence:** 4

**Summary:**

This paper addresses the evaluation tasks: Basic Spatial Relationships Understanding of direction, distance,and position in 2D space, in addition to the toy 2D space problem of Shortest Path Problem (SPP) and Traveling Salesman Problem (TSP). In experiments, they confirmed the effectiveness of tuning with the Sparkle-Instruct dataset with InternVL2-8B. It is also almost incredible but authors show that tuning with Sparkle-Instruct improves performance in out-of-domain (OOD) settings of COCO-spatial and GQA-spatial benchmarks.

**Strengths:**

1. Proposing a new dataset of 2D space reasoning that the current SoTA VLM models fail to perform well.
2. Training with the Sparkle-Instruct seems beneficial with InternVL2-8B, confirming the effectiveness both in the basic tasks, TSP and SPP.
3. Training with Sparkle-Instruct improves performance in out-of-domain settings of COCO-spatial and GQA-spatial benchmarks.

**Weaknesses:**

1. The experimental setting is a toy, lacking a concrete application of this benchmarking. We expect authors to clarify the benefit of this benchmark set and concrete applications.
2. The explanation of the novelty of this sparkle framework is limited. This framework seems to require the combination of abilities (e.g., direction, localization, distance) that are partially addressed in the existing benchmark sets.
3. Limited dataset size of 2000 images as of artificial dataset.
4. It is not fully-explained why Sparkle-Instruct improves performance even in OOD, although it is quite interesting and not intuitive. Indeed, this is totally unpredictable considering the limited dataset size and the sparkle framework is oriented for the artificial images. It is expected that authors detailedly explain how they performed their experiments in Section 4.2.3. It is strongly expected how the InternVL2 model is benefitted from Sparkle-Instruct for OOD of COCO-spatial and GQA-spatial with qualitative examples. It is also expected that authors explain the details of this for reproductivity.

**Questions:**

1.  I assumed that Sparkle-Instructs do not include real-world images. L367-L370 state that they use What’sUp, COCO-spatial, and GQA-spatial images for evaluation, but not for training purposes. If authors use some real-world images in the training, please clarify it.
2. Please further clarify why Sparkle-Instructs are effective even in OOD sets.
3. Why is the proposed dataset of Sparkle missing from Table.1? This makes the position of the proposed dataset unclear.
4. It is also curious why authors put the experiments with Qwen-VL-7B in Appendix, not in the main paper. Are there some explanations to do so?
5. SPP and TSP have explicit solvers that are not used in this paper. It is interesting if authors let models use some external tools as solvers.

---

### Official Review · Reviewer_oNwr · 2024-11-04

**Soundness:** 2
**Presentation:** 3
**Contribution:** 2
**Rating:** 3
**Confidence:** 4

**Summary:**

The paper explains how to improve the performance on composite spatial reasoning problems, specifically, SPP and TSP, with learning on three basic spatial capabilities, direction, distance and localization. The introduced framework, Sparkles, trains a model on synthetic data generating for three basic spatial capabilities.

**Strengths:**

The paper identifies three basic components of spatial reasoning and trains models to improve SPP and TSP task.
Apart from the synthetic SPP and TSP tasks, the Sparkles fine-tuned model also generalizes to some public spatial reasoning dataset.

**Weaknesses:**

- A very large portion of the paper is discussing performance on artificial tasks SPP and TSP, and the improvement is most on SPP. While looking at the synthetic data on basic spatial capabilities, direction, distance and localization, they are clearly subtasks of SPP and it's obvious fine-tuning on these tasks improves SPP (TSP may require more complex algorithms). I think the OOD generalization section is important to show the training on basic tasks *actually improves real VL spatial reasoning abilities*, not serving as auxiliary tasks for certain tasks. I'm sad there's only three short paragraphs for that. You should expand it and put in much more content in your next draft.
- Why we even need training? Prompting and in-context learning methods are not compared to. I think an easy way is just to insert three basic spatial reasoning questions as CoT before generating the final answer.
- In the OOD generalization test, why you choose What’s Up, COCO-Spatial and GQA-Spatial? This choice is weird, as in Table 1 there are datasets such as SpatialRGPT and QualSR requiring all three abilities you've trained on. It's important to understand if the training on all three basic tasks improves the most on these datasets vs. other datasets.
- Listed in Table 1, there are 5 basic abilities required in general for VL spatial reasoning, why you only choose to generate synthetic data and train on three of them? There should be consistent explanation and story for your choice.
- Missing ablations: as the main finding is not surprising, the experiments should be as comprehensive as possible. For example, Figure 6 should also have experiments with any combination of two basic tasks. The current figure does not answer the question why two basic tasks are not enough.

**Questions:**

See weeknesses.

---

### Official Review · Reviewer_f9Nx · 2024-11-04

**Soundness:** 2
**Presentation:** 3
**Contribution:** 2
**Rating:** 5
**Confidence:** 4

**Summary:**

This paper explores training Vision Language Models (VLMs) with enhanced spatial reasoning capabilities by focusing on three core spatial skills — direction comprehension, distance estimation, and localization. Sparkle, the proposed framework, aims to develop these foundational skills to improve VLM generalization on composite spatial tasks.

**Strengths:**

- The paper is well-motivated, breaking down spatial reasoning into foundational elements and underscoring the importance of direction, distance, and localization in visual language tasks. This structured approach effectively highlights the need for core spatial capabilities within VLMs.
- The authors present strong evidence of Sparkle’s impact, demonstrating notable increases in accuracy and generalization to unseen tasks. The positive results on real-world spatial benchmarks, such as What’s Up and COCO-spatial, reinforce the framework’s potential for enhancing generalizable spatial reasoning.

**Weaknesses:**

- This work only focuses on exploring 2D spatial reasoning capabilities within VLMs.
- Although Sparkle shows advancements in 2D spatial reasoning on benchmarks like What’s Up and COCO-spatial, it does not address perspective variations in real images. If the training images are all from straight-on views, this may restrict the model’s ability to generalize effectively in real-world applications where the perspectives vary.
- The fine-tuning is only performed on InternVL2-8B.

**Questions:**

- In Table 3, why does InternVL2-8B perform better than 26B without fine-tuning?

---

### Official Review · Reviewer_QL6U · 2024-11-05

**Soundness:** 2
**Presentation:** 3
**Contribution:** 2
**Rating:** 5
**Confidence:** 4

**Summary:**

The paper introduces SPARKLE, a framework designed to enhance the 2D spatial reasoning capabilities of Vision Language Models (VLMs). VLMs, despite their impressive performance in various tasks, struggle with spatial reasoning, particularly in composite spatial tasks like pathfinding. SPARKLE aims to improve these capabilities by focusing on three fundamental spatial reasoning skills: direction comprehension, distance estimation, and localization. The framework uses synthetic data generation and targeted supervision to create an instruction dataset for each capability, which is then used to fine-tune VLMs. The experiments show performance gains in both basic and composite spatial reasoning tasks, demonstrating the effectiveness of mastering basic spatial capabilities for enhancing composite spatial problem-solving.

**Strengths:**

1. Identifies key foundational spatial capabilities in spatial reasoning.
2. Experimental results demonstrate improved performance in both basic and composite spatial reasoning tasks.

**Weaknesses:**

1. The generalizability of synthetic data remains uncertain. It would be beneficial to test Sparkle on other open-source VLMs to assess whether performance gains extend beyond the primary model. The paper primarily focuses on the effectiveness of SPARKLE on the InternVL2-8B model. While the results are promising, the generalizability of these findings to other VLMs is not extensively tested. The synthetic data generated for training might be tailored to the characteristics of the model used in the experiments, and it is unclear how well these improvements would transfer to other open-source VLMs.
2. The paper does not provide a comprehensive evaluation across a diverse set of VLMs. Testing SPARKLE on a broader range of models could reveal its robustness and applicability across different architectures and training regimes. The paper notes only modest improvements in general spatial reasoning tasks. This suggests that while the framework is effective for spatially oriented problems, its impact on a wider array of visual tasks is less pronounced.
3. While fine-tuning in-domain improves performance, which is expected, there is only modest improvement on general spatial reasoning tasks. Testing Sparkle on more general visual tasks, rather than spatial reasoning-specific tasks, could reveal whether its performance holds across broader tasks. The paper does not extensively evaluate SPARKLE's performance on non-spatial reasoning tasks. It is unclear whether the enhancements in spatial reasoning translate to improvements in other visual tasks, such as object recognition or image captioning, which are also critical for VLMs. There is a risk that the model may overfit to the synthetic data used for fine-tuning, leading to less robust performance on real-world, diverse datasets that include a variety of spatial configurations not seen during training.

**Questions:**

Please refer to the weaknesses.

---

### Note · Authors · 2024-11-13

I have read and agree with the venue's withdrawal policy on behalf of myself and my co-authors.